

# Possibilistic response surfaces: incorporating fuzzy thresholds into bottom-up flood vulnerability analysis

Thibaut Lachaut[1] and Amaury Tilmant[1]

[1]Laval University, Québec, QC G1V 0A6, Canada

**Correspondence:** Thibaut Lachaut (thibaut.lachaut.1@ulaval.ca)

**Abstract.** Several alternatives have been proposed to shift the paradigms of water management under uncertainty from predictive to decision-centric. An often-mentioned tool is the stress-test response surface, mapping system performance to a large sample of future hydro-climatic conditions. Dividing this exposure space between acceptable and unacceptable states requires a criterion of acceptable performance defined by a threshold. In practice, however, stakeholders and decision-makers may be

confronted with ambiguous objectives for which the the acceptability threshold is not clearly defined (crisp). To accommodate such situations, this paper integrates fuzzy thresholds to the response surface tool. Such integration is not straightforward when response surfaces also have their own irreducible uncertainty, from the limited number of descriptors and the stochasticity of hydro-climatic conditions. Incorporating fuzzy thresholds therefore requires articulating uncertainties that are different in nature: the irreducible uncertainty of the response itself relative to the variables that describe change, and the ambiguity of the

acceptability threshold. We thus propose possibilistic surfaces to assess flood vulnerability with fuzzy acceptability thresholds. An adaptation of the logistic regression for fuzzy set theory combines the probability of acceptable outcome and the ambiguity of the acceptability criterion within a single possibility measure. We use the flood-prone reservoir system of the Upper Saint-François River Basin in Canada as a case study to illustrate the proposed approach. Results show how a fuzzy threshold can be quantitatively integrated when generating a response surface, and how ignoring it might lead to different decisions. This

study suggests that further theoretical development should link the decision-making under deep uncertainty framework with the existing experience of fuzzy set theory, notably for hydro-climatic vulnerability analysis.

## 1 Introduction

Uncertainty is a defining feature of water management - both science and practice. Not only does uncertainty justify the need for infrastructures, interventions and planning, but the way decisions are taken is also based on different interpretations of

uncertainty. The dominant paradigm has been to optimize investments or management plans according to the most probable future, assuming that the underlying processes are stationary. However, the stationary assumption has been contested as anthropogenic activities do affect the very processes that govern the water cycle (Milly et al., 2008).

Water resource planning approaches usually requires a prediction of those processes affecting both sides of the supply-demand relationship. In particular, the assessment of climate change impacts commonly relies on General Circulation Models

(GCM) that simulate future global climates based on assumptions about greenhouse gas concentrations in the atmosphere,



or more generally, the radiative forcing (Brown and Wilby, 2012, Weaver et al, 2013). Assumptions about greenhouse gas emissions, land cover or population are grouped under representative concentration pathways (RCP). Results from global simulations are translated into local hydro-climatic projections through a downscaling process. Hydrological modelling then translates climatic variables into run-off time series. Such an approach has its own limitations, however. $CO_2$ emission path-
ways depend on worldwide future policy choices which are not yet known nor even predictable. Moreover, climate models carry their own structural uncertainties, and so do the downscaling processes (Prudhomme et al., 2010, Mastrandrea et al., 2010, Kay, et al., 2013, Weaver et al., 2013, Kim et al., 2019). Besides, a discrete set of projections is not suited to find the hydro-climatic thresholds beyond which a system fails to reach its target (Culley et al., 2016). Such a risk assessment process is also increasingly unreliable with systems that operate with shorter time steps and extreme events, like flood control operations
(Knighton et al., 2017).

In the last 15 years there has thus been a widespread effort to find new paradigms to make decisions under deep uncertainty, notably through a greater focus on the decision process rather than on improving predictions (Lempert et al., 2006, Maier et al., 2016, Lempert, 2019). Switching to a robust or decision-centric paradigm always seeks to increase the sampling of hydro-climatic conditions, and relies on a sensitivity analysis of a water system to stressors rather than evaluating the consequences
of the most probable future and optimizing accordingly (Weaver et al., 2013). A consolidation of the field is proposed under the decision making under deep uncertainty (DMDU) denomination (Marchau et al., 2019).

One of the most common tools within the decision-centric framework is the response function or surface (Prudhomme et al., 2010, Brown et al., 2012, Culley et. al., 2016, Brown et al., 2019, Nazemi et al., 2020, Di Francesco et al., 2020). Through a stress-test, "bottom-up" approach, a water system is simulated for a large set of conditions representing possible evolutions of
some uncertain hydro-climatic variables (or stressors), establishing a relationship between such stressors and the performance of the system. When specifically addressing climate change, it corresponds to the reverse CIRF (Climate Impact Response Function, Fussel et al., 2003, Marcos-Garcia et al., 2020). Such an approach is sometimes called scenario-neutral (Prudhomme et al., 2010, Broderick et al., 2019) as it separates the system response from the likelihood of each scenario. Alternatives, like making new investments, changing management schemes, are compared through their respective performance outcome over a
whole range of possibilities (states of the world), or exposure space (Culley et al., 2016). The response surface can be used to measure an uncertainty horizon between a first estimate of the state of the world and an acceptability frontier (Info-gap decision theory, Ben-Haim, 2006). In the Decision Scaling approach (Brown et al., 2012, Brown et al., 2019) GCM projections can then be introduced as weights on the response surface to inform probabilities associated to climate states. GCMs can thus remain useful without conditioning the decision process, and their weights can be updated as uncertainty is resolved, resulting in a
refined estimate of the expected system outcome over the response surface without the need for new simulations of the water system. The intention shared within the overall decision-centric framework is to adapt classic risk assessment to the "death of stationarity" (Milly et al., 2008) while producing information more useful and engaging than a fully descriptive scenario approach (Weaver 2013). Response surfaces have been illustrated by many case studies (e.g. Nazemi et al., 2013, Turner et al., 2014, Whateley et al., 2014, Herman et al., 2015, Steinschneider et al., 2015, Spence et al., 2016, Pirttioja et al., 2019, Ray et





al., 2020), expanded to many-objectives or stakeholder systems (Poff et al., 2016; Culley et al., 2016, Kim et al., 2019) and
sometimes officially adopted in management processes (Moody and Brown, 2013, Weaver et al., 2013, Brown et al., 2019).

Although the response surface is a powerful and efficient tool to circumvent the problems brought by "top-down", GCM-
based assessments, the incorporation of such tools in actual decision-making processes to date remain relatively recent and
scarce (Guo et al., 2018). Moreover, many assumptions associated with the stress test approach can introduce additional uncer-
tainty.

One source can be the ambiguity of the user-defined acceptability thresholds (Maier et al., 2016). The stress-test approach
needs performance target values in order to separate the exposure space between acceptable and unacceptable domains. How-
ever such thresholds are often unclear or arbitrary (El-Baroudy and Simonovic, 2004). Recently, Hadjimichel et al. (2020)
performed a sensitivity analysis on the definition of binary acceptability thresholds for a large number of stakeholders in a deep
uncertainty framework, demonstrating its impact on decision making. Fuzzy set theory (Zadeh, 1965) provides an analytical
framework to characterize and manipulate stakeholders' ambiguity (Huynh et al., 2007). It has been extensively used in the
water domain (Tilmant et al., 2002, El-Baroudy and Simonovic, 2004, Qiu et al., 2018) in particular to solve multi-objective
decision-making problems (e.g. Jun et al., 2013). However, to the best of our knowledge, fuzzy set theory has not yet been
used to handle imprecise thresholds between satisfactory and unsatisfactory regions of a response surface. The very notion of
an arbitrary threshold defining success, like flood control reliability above 0.95, can be considered as a departure from a strictly
probabilistic framework and could justify a complementary possibilistic approach based on fuzzy sets (Dubois et al., 2004).
This paper therefore introduces the use of fuzzy acceptability thresholds when building a response surface for decision-centric
vulnerability assessment.

However, the internal uncertainty of the response surface hinders the direct application of a fuzzy threshold. The selected
stressor variables can only partially explain hydro-climatic uncertainties, and stochastic realizations introduce noise in the
resulting system performance. As such, performance is an expected value rather than a deterministic one, and that estimate
might underestimate real risks. Irreducible uncertainty usually requires adaptive management (Brown et al., 2011), but there is
interest in integrating estimates of uncertainty into the response surface tool.

Kay et al. (2014) proposed the use of uncertainty allowances that could vary depending on the response type and catchment.
More specifically, flood control systems operate on shorter time scales and are harder to assess over long term climate shifts
(Knighton et al. 2017), increasing uncertainty in flood response functions. Kim et al. (2018) show how the choice of a longer
modelling time scale (daily vs hourly) can lead to risk underestimation. The choice of weather generator used to generate
synthetic weather series in a scenario-neutral experiment can also lead to different results (Keller et al., 2019, Nazemi et al.,
2020). Steinschneider et al. (2015) and Whateley and Brown (2016) compare different sources of uncertainty in the response,
acknowledging the strong impacts of hydrological modelling and internal climate variability compared to long term climate
uncertainty. Testing a limited number of stressors as explanatory variables therefore leads to a response function that returns
imprecise performance estimates. Quin et al. (2018), Kim et al. (2019), Lamontagne et al. (2019), Hadjimichael et al. (2020),
and Marcos-Garcia et al. (2020) use a logistic regression to divide the exposure space based on probability of success (often as
a first step in a scenario-discovery approach). Tanner et al. (2019) do so with a Bayesian belief network model.





Such internal uncertainty to the response function challenges the introduction of fuzzy thresholds, as the separation of the exposure space in acceptable and unacceptable regions is not obvious even with a binary definition.

Through a possibilistic approach, the present study combines two different types of uncertainty: the fuzziness of acceptability thresholds and the irreducible uncertainty of the response surface. The rationale of the paper is further developed in section 2, followed by the incorporation of a fuzzy acceptability threshold to the logistic regression, which has already been used to

handle the response uncertainty. A case study is presented in section 3, a flood-prone reservoir system in southern Québec, Canada. Results are presented in part 4, followed by a discussion on the merits and limitations of the proposed method.

## 2 Methods

### 2.1 Rationale

#### 2.1.1 Uncertain response function

We first consider how a limited set of variables leads to an inherently uncertain response function, and how it relates to the partition of the exposure space and the decision process.

A stress test consists in assessing the performance of a system for a large enough number of situations, in order to identify which of these situations lead to an unacceptable performance.

We first define how success, failure, performance and acceptability are used in this study. Success or failure are a state of

the system at a given time step. For example failure can be defined by a streamflow exceeding a threshold at any moment, characterizing a state of flooding. Inspired from Hashimoto et al. (1982), common performance indicators of a water system are statistical measures of the frequency, amplitude or duration of failures, aggregated over a certain time period. For example, the reliability of a flood control system can be measured as the proportion of a given period (frequency) where no flooding happens.

While success/failure define the state of the system for a single time step, acceptable/unacceptable define its behavior over a time period. When performing a stress-test of a system, the criterion for acceptable/unacceptable outcomes is usually defined by performance satisfying a threshold $\theta$, for example reliability above 0.95 over a given period can define an acceptable outcome.

A stress-test maps the performance $R$ on a response surface, to a limited number of descriptive variables or stressors $x_i$. Each coordinate, or state of the world, is a combination of specific values taken by such stressors. The stress-test aims to

delineate the subsets $A$ and $D$ of acceptable and unacceptable outcomes (Fig. 1). Alternative options (management rules, infrastructure design) can be ranked based on the respective size of subsets A and D. The more states of the world lead to acceptable outcomes, the more robust an option is, thus the more preferable in this approach.



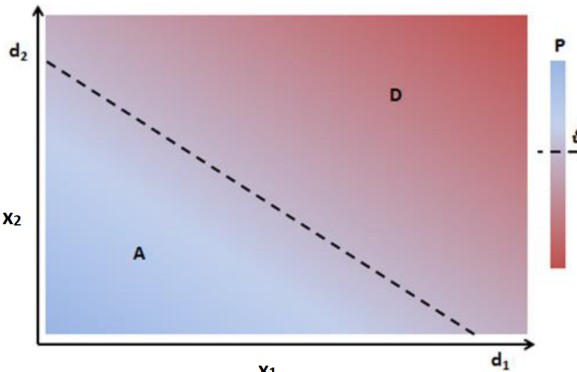

**Figure 1.** Concept of the response surface as a stress-test with descriptive variables $(x_1, x_2)$. Acceptable and unacceptable regions are defined by a threshold $\theta$ over performance $R$.

The descriptive variables or stressors, like the mean flow, the peak flow, or temporal autocorrelations, are aggregations of the time series that are the inputs of a water system model. Because a limited number of descriptors do not capture all possible fluctuations of a time series, a term of irreducible uncertainty remains. The response surface, $R$, is then given by:

$$R = g(x_1, x_2...) + \epsilon \tag{1}$$

In a risk-averse approach, the objective is to find the range of unacceptable outcomes, the space over which a system fails to satisfy an acceptability threshold $\theta$. With 2 variables, this space is the set of solutions $D = (x_1^*, x_2^*)$ to the inequality $p < \theta$, so

$$g(x_1, x_2) + \epsilon < \theta \tag{2}$$

Simplifying the response surface by e.g. its average estimate can thus under-estimate the unacceptability domain. Irreducible uncertainty can be addressed through adaptive management (Brown et al., 2011), uncertainty allowances (Kay et al., 2014), and extensive Monte-Carlo sampling (Whateley and Brown, 2016). If possible though, it can be convenient to directly integrate information about the remaining uncertainty within the response surface itself. It can be represented through a transition zone between success and failure domains with a logistic regression.

### 2.1.2 Fuzzy acceptability thresholds

The acceptability criterion based on a threshold $\theta$ defines the set of acceptable outcomes. It is a subjective or arbitrary opinion from stakeholders or decision makers to attribute a normative value to a certain performance level. The vast majority of the studies reported in the literature assume that the threshold between satisfactory and unsatisfactory outcomes is crisp (Brown et al., 2012, Culley et al., 2016, Kim et al., 2019). As such a threshold shapes directly the partition of the response function, with a crisp value the exposure space can be subdivided in only two sub-spaces: acceptable versus unacceptable.



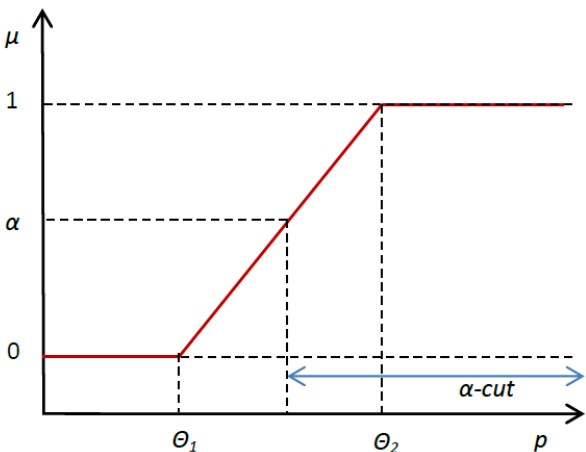

**Figure 2.** Concept for a fuzzy set of acceptable outcomes $A_\mu$ over performance $R$.

The very existence of a threshold is the basis of satisficing behaviors (Simon, 1955) that differ from utility maximizing behaviors as coined by Von Neumann and Morgenstern (1944). In practice however, there might be situations whereby the water manager is unable (or unwilling) to provide a crisp, well-defined threshold, or when such threshold is disagreed upon by stakeholders. For example, when controlling water levels in a reservoir to prevent floods, the operator can handle certain

tolerances above the maximum desired level. Of course, the greater the deviation from the desired level, the less acceptable it becomes.

Mathematically, fuzzy sets theory handles imprecisely-defined or ambiguous quantities. Introduced by Zadeh in 1965, fuzzy sets theory has become a common tool in decision making analysis or computational sciences when non-probabilistic uncertainty stemming from ambiguity or vagueness must be considered (Yu et al., 2002). In our case, fuzzy sets theory allows us to

introduce vagueness in target-based decision making, without forsaking a target-based model in favor of an unbounded maximizing behavior (although a fuzzy target can also be seen as a generalization of both maximizing and satisficing behaviors – see Castagnoli and LiCalzi, 1996, and Huynh et al., 2007).

We consider here the case where such a threshold $\theta$ may not be precisely defined by stakeholders but can take many subjective qualifications from acceptable to unbearable, hence relaxing (without fully removing) the arbitrary condition of satisfying a

crisp value. A fuzzy set $A_\mu$ of acceptable states therefore qualifies the performance $R$ with a membership value comprised





between 0 and 1. The membership function $\mu$ associated to the fuzzy set $A$ describes the degree to which any value of $R$ more or less belongs to $A$ (Figure 2, eq 3).

$$\begin{cases} \mu(R) = 0 & R < \theta_1 \\ 0 < \mu(R) < 1 & \theta_1 \geq R < \theta_2 \\ \mu(R) = 1 & R \geq \theta_2 \end{cases} \qquad (3)$$

When a threshold corresponds to a fuzzy set, it means that there is a transition zone between acceptable and unacceptable

outcomes where intermediate levels of membership exist. Conversely, another interpretation is that the membership function is the distribution of the *possibilities* (Zadeh 1978, Dubois and Prade, 1988) that any given performance $R$ represents an acceptable outcome.

An $\alpha$-cut $A_\alpha$ is the crisp set over $A_\mu$ for which the membership degree to $A_\mu$ is equal or above $\alpha$. The largest $\alpha$-cut is called the *support* of the fuzzy set $A_\mu$ ($R \geq \theta_1$). The smallest $\alpha$-cut is the *core* of the fuzzy set ($R \geq \theta_2$).

$A_\alpha = \{R \in A_\mu \mid \mu(R) \geq \alpha\}$          (4)

A fuzzy definition of acceptability is not only a way to accommodate ambiguity as a stakeholder-based constraint, it can also alter the outcome of the analysis. Theoretically, it can happen when the slope of a response as function of stressors is different for the compared alternatives (infrastructure investments, management rules...), as illustrated in Fig. 3 for a single stressor variable. Rule 2 has a larger region of success with a crisp threshold, but the result is mixed with a fuzzy definition of

acceptability. In that case, a trade-off appears between minimizing a loss of any sort (i.e. any type of flooding), and minimizing the maximum loss (min-max).

For example, in Quinn et al. (2017a), an attempt at reducing flooding most of the time leads to worse results under extreme events. Hadjimichael et al. (2020) perform a sensitivity analysis on binary acceptability thresholds, show the impact such definition has on the outcome of a vulnerability assessment. Criterion ambiguity could lead to similar effects: the preferred

option might not be the same depending on the value of the threshold, and in the present study depending on the degree of acceptability.

## 2.2    Combination of fuzzy thresholds and uncertain response function

When incorporating a fuzzy threshold, the challenge is to combine two different sources of uncertainty described in section 2.1: the uncertainty of the response itself relative to the variables that describe change, and the ambiguity of the acceptability

threshold. An approximated fuzzy-random logistic regression is proposed in order to integrate both.

### 2.2.1    Approximation of a fuzzy-random logistic regression

As the goal of the response surface is to divide the exposure space between acceptable and unacceptable outcomes, the value associated to any combination of variables can be either 0 or 1 if a specific acceptability threshold $\theta$ is reached or not. As seen



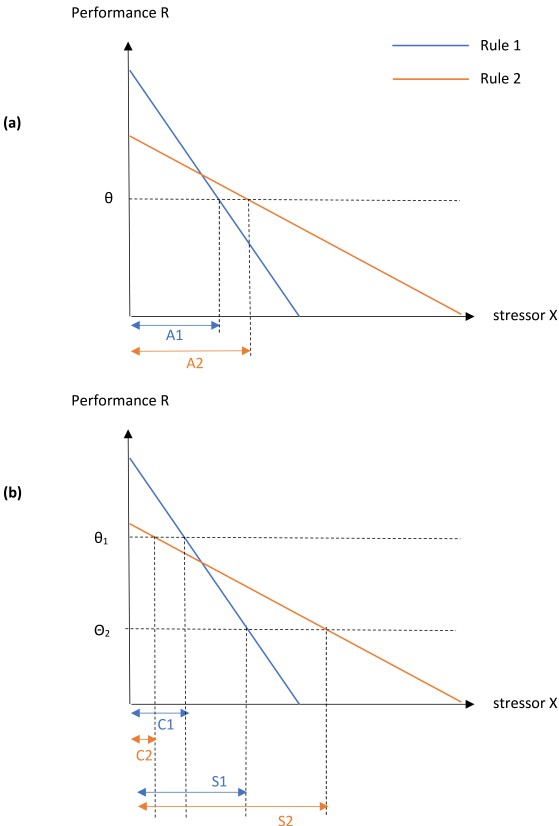

**Figure 3.** The case of different performance slopes, as function of a single stressor $X$. (a) With a crisp threshold $\theta$, rule 2 has a larger acceptable region A2. (b) With a fuzzy threshold $(\theta_1, \theta_2)$, the fuzzy set of acceptable outcomes over performance $R$ has a core ($R \geq \theta_1$ , where acceptability $\mu = 1$) and a support ($R \geq \theta_2$, where $\mu \geq 0$), to which respective regions C and S are associated. Rule 2 has a larger "at least partial" acceptability domain S2, but a smaller "full" acceptability domain C2, than Rule 1.

in section 2.1, an intrinsic uncertainty remains in response surfaces. Quin et al. (2018), Kim et al. (2019), Lamontagne et al.
(2019), Hadjimichael et al. (2020), and Marcos-Garcia et al. (2020) use a logistic regression to divide the exposure space based on probability of success. The logistic regression is used to explain a binary outcome from independent variables $(x_1, x_2)$, and returns a probability of success $\pi$ :

$$\pi_\theta = \frac{1}{1 + \exp\left(-\left(\beta_0 + \beta_1 x_1 + \beta_2 x_2 + \dots\right)\right)} \tag{5}$$

$\quad \pi_\theta(x_1, x_2) = P(R \geq \theta) \tag{6}$

where $x_i$ are the defining variables of the exposure space and $\beta_i$ the regression coefficients. The logistic response surface therefore provides the probability $\pi$ of meeting the threshold $\theta$ over the $(x_1, x_2)$ exposure space. The logistic regression also



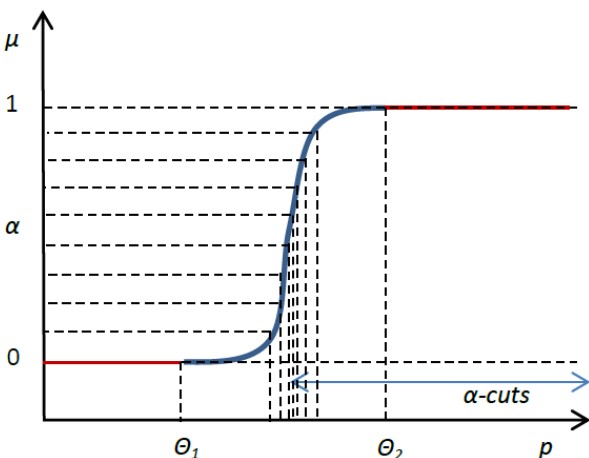

**Figure 4.** Concept for $\alpha$-cut sampling, sigmoid function

has its own uncertainty but it is not considered here. While the response surface considers a range of states of the world without knowing their probability of occurrence, the logistic regression still provides a conditional probability of acceptable outcome

once a given state of the world is reached. Partitions of the space between acceptable and unacceptable sub-spaces, that can be defined as $\pi - cuts$, are now relative to a specific probability of success $\pi^*$ taken by $\pi_\theta$:

$$S_{\pi^*} = \{x_1, x_2 \mid \pi\ (x_1, x_2) \geq \pi^*\} \tag{7}$$

By considering the domain of successful outcomes as a fuzzy set, we introduce a layer of uncertainty that is different in nature from the irreducible hydro-climatic uncertainty. While the logistic regression returns a *probability* of surpassing any

given acceptability threshold for a combination of variables (eq. 5 and 6), the fuzzy set of success returns the *possibility* of any such performance value being actually considered as a success (eq. 7).

Fuzzy regression models, including fuzzy logistic regression (e.g. Pourahmad et al., 2011, Namdari et al., 2014) replace probabilities by fuzzy numbers; they usually do not combine them. Fuzzy probabilities (Zadeh, 1984) are considered within the so-called fuzzy random regression field, however no fuzzy random logistic regression seems to have been developed to date

(see Chukhrova and Johannssen, 2019, for a review of the fuzzy regression field).

Here we use a discretised approximation of a fuzzy random logistic regression based on $\alpha$-cuts. As illustrated in Fig. 2 and Fig. 4, a fuzzy set $A_\mu$ can be decomposed in alpha-cuts. Each $\alpha$-cut is a crisp set, and the values belonging to an alpha-cut also belong to the fuzzy set $A_\mu$ with a membership degree equal or above $\alpha$.



Therefore, any crisp set of acceptable outcomes $A$, defined by a single threshold $\theta$, is also an $\alpha$-cut of the fuzzy set of success
$A_\mu$. Then a single logistic regression for any success threshold $\theta$ is also the probability of belonging to the $\alpha$-cut of the fuzzy
set of success defined by $\theta$:

$$\pi_\theta(x_1, x_2) = P(R \in A_\alpha) = P(R \in A_\mu \mid \mu(R) \geq \alpha) \tag{8}$$

with $\alpha = \mu(\theta)$.

Following the interpretation of Huynh et al. (2007), the overall possibility $\Pi$ of the random variable $R$ belonging to the fuzzy
set $A_\mu$ can be given by the integral over $\alpha$ of the probabilities of success defined at every $\alpha$-cut.

$$\Pi(x_1, x_2) = P(R \in A_\mu) = \int_0^1 P(R \in A_\mu \mid \mu(R) \geq \alpha)\, d\alpha \tag{9}$$

And thus

$$\Pi(x_1, x_2) = \int_0^1 \pi_{\mu^{-1}(\alpha)}(x_1, x_2)\, d\alpha \tag{10}$$

The approximated logistic regression for a fuzzy set of success is therefore the average of the logistic regressions for all the
associated $\alpha$-cuts. With a uniform discretization of 10 alpha levels, the spacing of every $\alpha$-cut, defined with $\theta = \mu^{-1}(\alpha)$, relies
on the shape of the membership function. A linear shape of $\mu(R)$ leads to a uniform sampling of the $\alpha$-cuts, while a sigmoid
error function leads to a Gaussian sampling of $\alpha$-cuts centered on $\theta^* = \mu^{-1}(0.5)$ (Fig. 4).

## 3 Application

A two-reservoir system in eastern Canada is used as a case study to illustrate the applicability of the possibilistic response
surfaces.

### 3.1 Upper Saint-François River Basin features

The Upper Saint-François River Basin (USFRB) is located in the province of Québec, Canada. The selected gauging point,
near the agglomeration of Weedon, drains an area of 2940 km$^2$ with an average annual flow of 2.1 billion cubic meters. The
system (Fig. 5) involves the Saint François River, controlled by two reservoirs Lake Saint-François and Lake Aylmer with a
combined storage capacity of 941 million cubic meters, and the uncontrolled affluent Saumon River.

Both reservoirs are managed by the Québec Water Agency which is part of the Ministry of Environment (Ministère de
l'Environnement et de la Lutte contre les changements climatiques - MELCC). The main operational objectives are: (i) to
protect the municipality of Weedon and several residential areas around the lakes from floods, (ii) to ensure minimum river





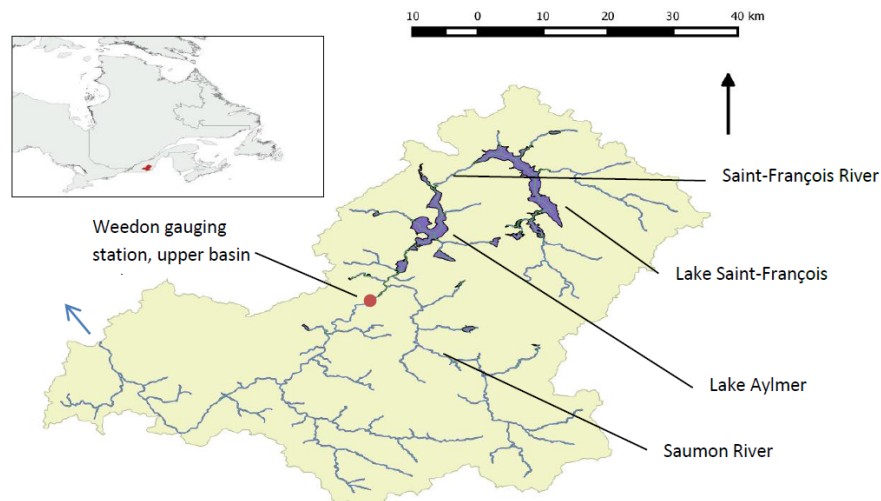

**Figure 5.** Layout of the Upper Saint-François River Basin, Québec, Canada.

discharges and water levels in the lakes to preserve aquatic ecosystems, (iii) to regulate the floods for downstream power plants;
and (iv) to maintain desired water levels in the lakes for recreational uses during the summer.

This multipurpose reservoir system thus follows a refill-drawdown cycle accordingly. With a snowmelt-dominated flow regime, the reservoirs are emptied in winter, filled during the spring and kept at constant pool elevation during the summer.

## 3.2   Inflow time series

At the request of system operators we use hydrologic stressors instead of climatic ones. Other authors have also used hydrologic
stressors, see e.g. Nazemi et al., 2013, Borgomeo et al., 2015, Herman et al., 2016, Zeng et al., 2017, or Nazemi et al., 2020.
Readily available inflow time series from GCM weather projections are used to generate additional synthetic streamflow series
as in Vormoor et al., 2017. Results are then directly plotted on the exposure space according to their own $(x_1, x_2)$ coordinates.
Such a method seeks to make a greater use of hydro-climatic future scenarios when many are already available, to obtain a
higher diversity of synthetic times series (based on different GCM simulations, RCP scenarios and downscaling techniques).
We first describe the initially available time series, then how they are perturbed and reused to create synthetic time series.

Historical daily measurements are available for the 2000-2014 period (MELCC, 2018). They include lakes inflows, levels
and reservoir releases, and river discharges from the tributary and at the basin outlet.

Streamflow scenarios are provided by MELCC through the Quebec Water Atlas 2015 (CEHQ, 2015, MELCC, 2018). Those
hydrologic projections are based on climatic projections from the Natural Resources Canada data base of GCM simulations
(CMIP5, Hydro-climatic Atlas, 2015) that were downscaled by the Québec Water Agency. A set of 501 time series was made
available, spanning 30 years of daily inflows. The set contains 135 scenarios for a 1971-2000 reference period; and 366
scenarios for the 2041-2070 period. The 366 scenarios are based on 122 GCM projections, from which 3 different downscaling





techniques were applied: without bias correction, with quantile mapping or with delta quantile mapping (based on Mpelasoka and Chiew, 2009). In order to obtain the largest degree of variability, and find as many failure configurations as possible, all

501 time series are used indistinctively, first perturbed to increase the sample, then used as input for the synthetic time series generation.

In order to expand the sample of the exposure space and explore less favorable conditions, the perturbation of available inflows is performed by either modifying the average annual flow, the dispersion of daily flows, or both. To increase the range of tested inflow volumes, a single change factor is applied in the first case, arbitrarily increasing all flow values at every time

step by 50%. To perturb the dispersion, a varying factor multiplies flow values depending on their rank in the series distribution (factor 1 for the lowest, factor 1.5 for the highest flow). There are then 4 categories of perturbation: volume only, dispersion, volume and dispersion, and none.

This expanded set of time series is then used as input of the synthetic generator. The generator is the Kirsh-Nowak method (Nowak et al., 2010, Kirsh et al., 2013), made available online as Matlab code by Quinn et al. (2017b), employed e.g. in Quinn

et al., 2017a. Each synthetic generation is performed twice for each available time series. We then get 501 (initial tie series) ×4 (different perturbations) ×2 (synthetic realizations) = 4008 synthetic time series, each containing 30 years of daily river discharges.

$$G = \frac{1}{N}\left(N + 1 - 2\frac{\sum_{i=1}^{N}(N+1-i)\,q_i}{\sum_{i=1}^{N}q_i}\right) \qquad (11)$$

Similarly to other stress-test studies that generate inflow instead of climate time series (Feng et al., 2017), the selected

driving variables (axes x and y of the response function) are the total annual inflow volume and a measure of the intra-annual variability of streamflow. The intra-annual variability is here measured with the dispersion coefficient G, a measure also known as Gini coefficient in economics but employed too in hydrology (Masaki et al., 2014). It is similar to the coefficient of variation used in e.g. Nazemi et al. (2020) but bound between 0 and 1, which offers convenient interpretation: at G=0 all daily discharges in a year are equal, if G=1 the entire yearly run-off happens in a single day. Like the variation coefficient it allows for a second

variable statistically independent of the total annual run-off volume. Here $q_i$ are the ordered daily discharges of a given year, N=365 days.

### 3.3  Simulation and response surface

The model is built with HEC-ResSim, the Reservoir System Simulation software developed by the US Army Corps of Engineers (Klipsch and Hurst, 2007). It relies on a network of elements representing the physical system (reservoirs, junctions,

routing reaches), as well as the sets of operating rules. HEC-ResSim replicates the decision-making process applied to many actual reservoirs through a rule-based modeling of operational constraints and targets.

Hydrologic inputs consist of 30 years long, daily river discharges for each sub-basin. The main outputs are daily water levels in lakes, reservoir releases, as well as the discharges at the outlet. A complementary Jython routine is developed in order to





run HEC-ResSim in a loop to systematically load a large set of different hydro-climatic scenarios. Dam characteristics and

operational rules were provided by the Québec Water Agency (MELCC, 2018).

The model is developed with a first set of operating rules (rule 1) expected to mimic the current operation of the system. It reproduces measured daily releases over the 2000-2014 period. 4008 simulations are then run, each taking an input of synthetic daily flow series spanning 30 years. In order to increase the density of the un-gridded exposure space sampling, results are divided in 5 years periods. Such decomposition is deemed acceptable based on the reservoir system, which storage

capacity is designed for seasonal regulation, not multi-year, mitigating the effects of boundary conditions. It leads to a sample of 24'048 points, each one representing a five-year simulation.

Although the operating rules were designed taking into account all operating objectives, the present study focuses on the flood control performance $R$. More specifically, it is the reliability (Hashimoto et al., 1982) of the system keeping the river discharge at Weedon below 300 m$^3$s$^{-1}$. Mathematically, if $F(t)$ is the state of flooding at time step $t$, then $R$ is given by:

$$F(t) = \begin{cases} 0 \ \ if \ \ Q(t) \le 300 \\ 1 \ \ if \ \ Q(t) > 300 \end{cases} \tag{12}$$

$$R = 1 - \frac{1}{T}\sum_{t=1}^{T} F(t) \tag{13}$$

The response function is built by representing performance $R$ as a function of the selected inflow characteristics (yearly volume and dispersion). We consider the case where the threshold between acceptable and unacceptable performance is not

clearly defined, but is bounded between $\theta_1 = 0.93$ and $\theta_2 = 0.97$.

The separation of the exposure space between acceptable and unacceptable regions is calculated following section 2, combining a logistic regression with a fuzzy acceptability domain, its support being [0.93, 1] and its core [0.97,1]. Consequently, any given performance value $R$ has a membership degree of 0 for $R < 0.93$, and equal to 1 for $R \ge 0.97$. A counterfactual exercise is also run with a crisp threshold $\theta^*$=0.95, where the ambiguity is ignored and only the median between bounds is

selected. Performance is also calculated for a sub-set of GCM-based projections deemed more trustworthy by the CEHQ (with quantile mapping downscaling for the 2041-2070 period), each one divided by 5-years periods.

These rules are compared to an instrumental set rule 2 which slightly alters the anticipation and emergency release algorithm of the reservoirs.

The simulation is first run with 122 of the original time series made available by the Québec Water Agency. These are the

bias-corrected rainfall / run-off simulations considered as the most reliable for scenario-driven adaptation plans, corresponding to different radiative forcing scenarios. Taken by 5 year periods (thus 610 time series), all lead to flood control reliabilities superior to 0.97, above any considered acceptability threshold. So both rule sets are considered successful in all these time series.

Simulations are then run for the much larger and diverse sample of 4008 synthetic time series. The performance R, measured

as the reliability of flood control, is evaluated for each 5-years period contained in the 4008 simulations of 30 years (24'048





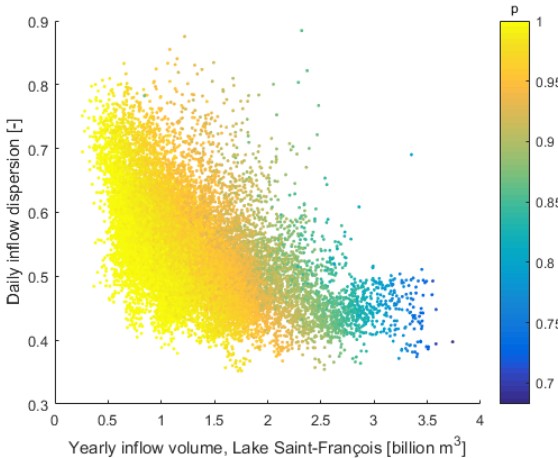

**Figure 6.** Response surface (rule 1). Performance $R$ : flood control reliability

evaluations). The color scale represents the performance (reliability $R$ against floods) in Fig. 6 for each 5 year time series. The axes are the stressors $x_1, x_2$: the average annual inflow volume at Lake Saint-François and the dispersion (or Gini coefficient) of the daily inflows. The response shows considerable noise, although a north-east / south-west anisotropy or gradient can be visually noticed.

## 4   Results

An acceptability value $\mu$ is then associated to each dot (time series) in the sample depending on the value of performance (reliability) $R$ (Fig. 7.a). The acceptability value $\mu$ is the membership degree of $R$ to the fuzzy set of acceptable outcomes, with [0.93, 1] as support and [0.97, 1] as core (as in Fig. 2). The sample of simulations thus leads to acceptability values between 0 and 1 in Fig. 7a.

To solve the problem of combining two uncertainties that are different in nature (probability of to meet a threshold vs possibility that this threshold is acceptable), the aggregated logistic regression presented in section 2.2.1. is performed for the fuzzy outcomes, thus proposing a continuous mapping for a case where the outcomes are not available as binary categories. The logistic regression is performed 10 times for 10 $\alpha$-cuts corresponding to a uniform sampling of $\alpha$-levels. The aggregated logistic regression at every coordinate is the average of the 10 logistic regressions, each one considering a single $\alpha$-cut as the crisp set over $R$ that defines acceptable outcomes.

It provides at each coordinate of the exposure space (or state of the world) a possibility value $\Pi$ of the outcome (reliability against floods) being deemed as acceptable given the realization of the state of the world. This – conditional – possibility measure expresses both the ambiguity of the acceptability criterion, and the probability of an acceptable outcome at any location on the response surface. The surface can be divided in acceptable and unacceptable regions (Fig. 7) based on any desired level of possibility ($\Pi$-cut).





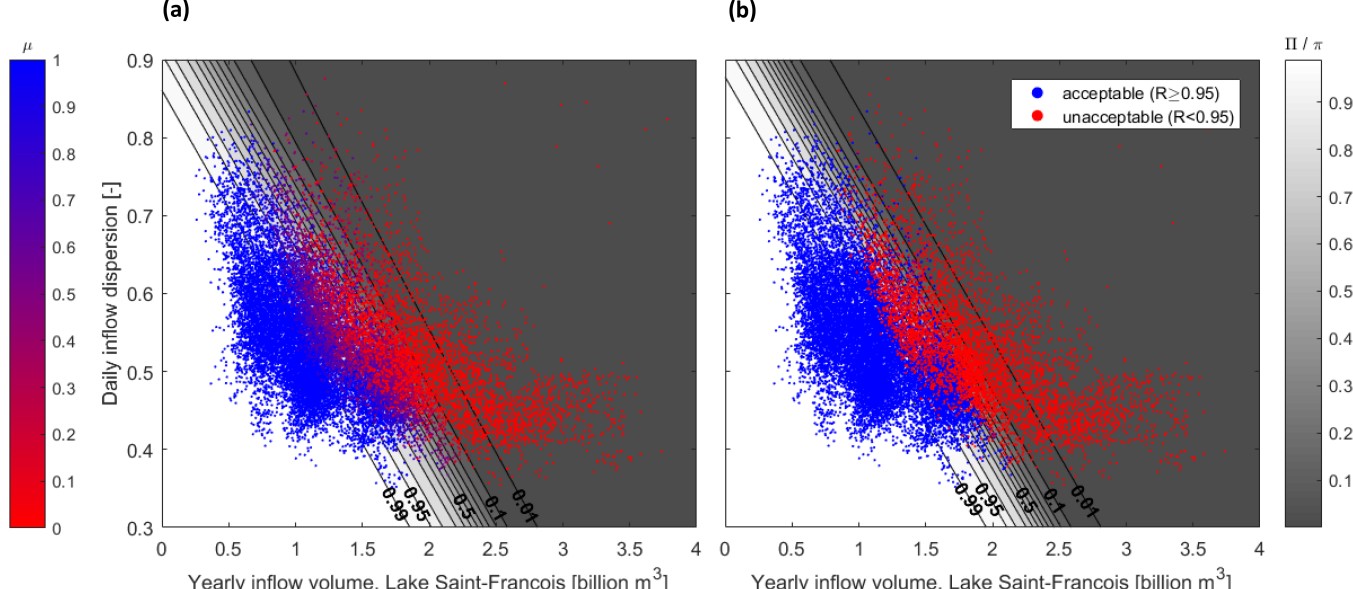

**Figure 7.** Acceptability of sampled outcomes and logistic regressions (a) Fuzzy outcomes and possibility of acceptable performance $\Pi$ (b) Binary outcomes and probability of acceptable performance $\pi$

As a counterfactual, we also compute a an alternative where the ambiguity of the threshold is ignored with the response surface converted into binary outcomes – acceptable or unacceptable frequency of flooding – based on a median crisp threshold of 0.95 (Fig. 7b). A simple logistic regression is performed for the counterfactual binary outcomes, leading to a probability $\pi$ of acceptable outcome.

340 The approximation was done with the Matlabs® function *mnrfit*. The McFadden pseudo $R^2$ of the median threshold logistic regression is 0.7531. The relation between explanatory variables is kept linear, as introducing an interaction term only increased the pseudo $R^2$ to 0.7562.

 $\Pi$-cuts producing frontiers between acceptability regions can be contrasted with the mapping of time series from GCM projections on the response surface (Fig. 8). While all these downscaled time series lead to fully acceptable performances,

345 showing reliability values above 0.97, their coordinates, thus their corresponding state of the world, can still fall below a $\Pi$-cut.

 This is because for any of these projections, the evaluated sequence is one realization of those conditions $x_1$, $x_2$. Assuming the logistic regression model is accurate, with possibility 1-$\Pi$, alternative realizations of those conditions may not be seen as satisfactory. A scenario sharing the same properties $x_1$, $x_2$ - yearly inflow volume and daily inflow dispersion – with a satisfactory GCM projection could still lead to unacceptable frequency of flooding if its possibility of acceptable outcome $\Pi$ is

350 below the acceptable level.

 Respectively, the binary counterfactual model (Fig. 8b), provides a degree of probability $\pi$ of an unacceptable outcome for the same state of the world, as in the previous studies using the logistic regression. Fig 8. illustrates the straightforward





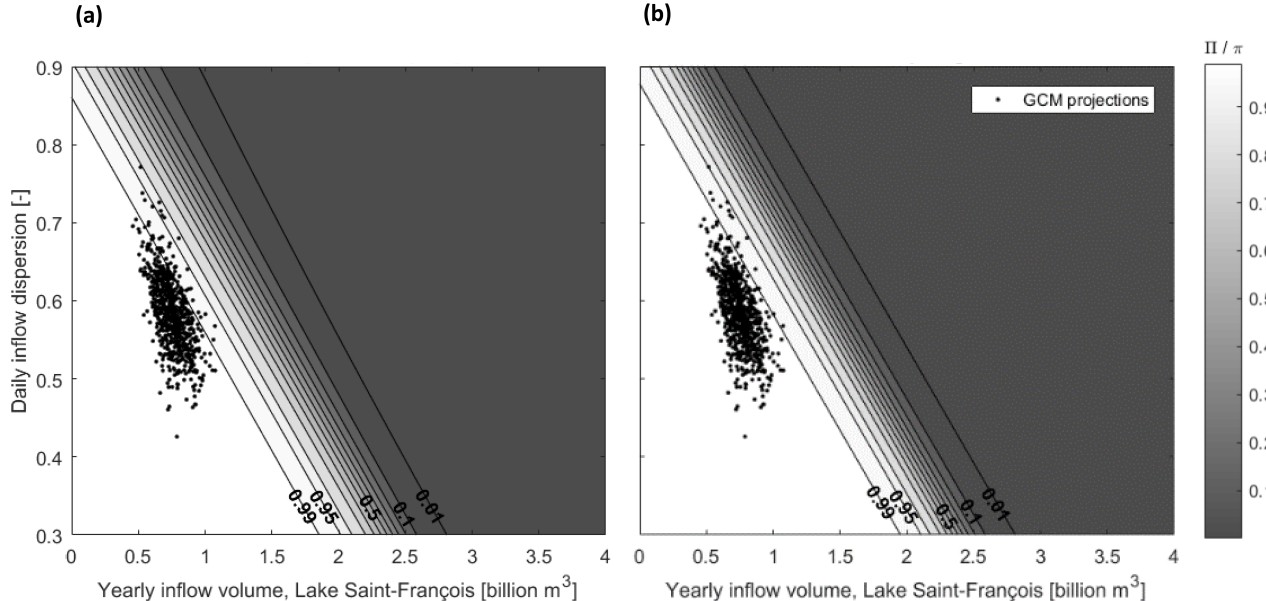

**Figure 8.** Logistic regressions and GCM projections (a) for fuzzy outcomes (b) for binary outcomes

difference between adapting to the ambiguity of the acceptability criterion (Fig. 8a) and ignoring it (Fig. 8b). For any given state of the world (coordinate) $x_1$, $x_2$, the aggregated logistic regression not only considers the probability of a realization leading

to a certain performance, but also the possibility that such performance, the reliability against floods, would be considered an acceptable outcome. Accepting a fuzzy acceptability criterion thus mechanically widens the range of the continuous transition resulting from the logistic regressions. A state of the world with a 100% probability of meeting a 0.95 reliability threshold might still have a possibility of this threshold not being *actually* accepted.

Such differences are more noticeable in this case study when using GCM projections as ex-post weights. With a fuzzy target,

46 projections (6.3%) fall out of the 0.99 Π-cut, i.e. the sub-space where the possibility of acceptable outcome is at least 0.99. Said otherwise, there is a possibility of at least 0.01 that a realization leading to the same state of the world (coordinates) would produce an flooding pattern considered as unacceptable, according to the aggregated logistic regression. With a crisp simplification, thus with less information, 17 projections (2.3%) fall out of the 0.99 $\pi$-cut. There is a probability of at least 0.01 that a realization leading to the same state of the world would produce an unacceptable outcome.

These frontiers are specific to a reservoir operation rule. When using a stress-test with a response surface, alternative rules are compared based on the relative position of the frontiers between acceptable and unacceptable regions. Two of them are here compared: an approximation of current reservoir operations (rule 1) and an alternative, instrumental set of rules (rule 2). Fig. 9 compares the two rules based on selected Π-cuts (Fig. 9a). The counterfactual calculation with binary outcomes and $\pi$-cuts is also provided (Fig. 9b).





Fig. 9 shows a situation partially similar to the theoretical situation of Fig. 2, where the relative advantage of each rule depends on the location on the exposure space and the preferred level of possibility. With a fuzzy acceptability criterion (Fig. 9a), rule 2 is preferred to rule 1 for high possibility of acceptable outcome ($\Pi \geq 0.95$), because the region defined by this frontier for rule 2 is larger than for rule 1. It means that rule 2 leads to acceptable outcomes in a larger range of states of the world than rule 1. However, for low possibilities of acceptable outcome (below 0.05), the comparison depends on the stressors $x_1$, $x_2$.

Rule 2 is preferred for very high daily inflow dispersion (or Gini coefficient, y axis) but moderate yearly inflow (x axis), while rule 1 is preferred for low dispersion and very high yearly inflows (again assuming the logistic regression model is accurate).

Using a counterfactual with binary outcomes (Fig 9b), and thus frontiers defined only by probabilities, modifies the above results. While rule 2 remains preferable for high probabilities of acceptable outcomes, it becomes worse than rule 1 for low probability-cuts, this time independently of the location on the exposure space.

If the decision-makers choose to use GCM projections as ex-post weights, the preference order for low possibility levels becomes less important. It narrows the relevance of the exposure space to the vicinity of the projections, thus to high possibilities of acceptable outcomes. In this case, 8 scenarios for rule 2 (1.1%) fall below the 0.99 $\Pi$-cut (meaning other realizations have a 0.01 possibility of unacceptable outcome), against 46 scenarios for rule 1 (6.3%). Again, all GCM-based scenario lead to fully acceptable outcomes ($R \geq 0.97$). Rule 2 would then be preferred to rule 1, but there would still be a possibility of unacceptable

outcome superior to 0.01 with this rule, for the same states of the world (coordinates $x_1$, $x_2$) sampled by the GCM-based scenarios.

Using only binary outcomes, thus only probability-based frontiers, produces a slightly different result. Rule 2 is not only preferred in the vicinity of the GCM projections, but also no such projection falls below the 0.99 $\pi$-cut. The probability of unacceptable outcome is thus less than 0.01 at the vicinity of any GCM projection. Based on such projections, rule 2 would be

adopted with less reservations with a binary model than with a fuzzy model.

## 5    Discussion

By itself, a stress-test approach can be seen as a departure from a probabilistic framework towards a possibilistic one. The stress-test of a water system, through a response surface, asks which states of the world could possibly lead to an unacceptable outcome, instead of evaluating the system performance with a given probability.

In this paper, we consider that the threshold employed to define acceptable outcomes might be ambiguous or contentious. The fuzzy or possibilistic framework (Zadeh 1965, 1978; Dubois and Prade, 1988), often used in decision-making analysis, provides the analytical tools to incorporate an uncertainty that is not probabilistic, the ambiguity of a decision threshold, within the increasingly popular stress-test tool.

Applying a fuzzy threshold would be straightforward for a deterministic response surface, each performance value on the

exposure space being mapped to a degree of acceptability between 0 and 1. This study explores how to solve the problem of combining a fuzzy definition of acceptability with the remaining hydro-climatic uncertainty of the response surface itself. The two sources of uncertainty are different in nature: one applies to the performance of the system, the other to the qualification





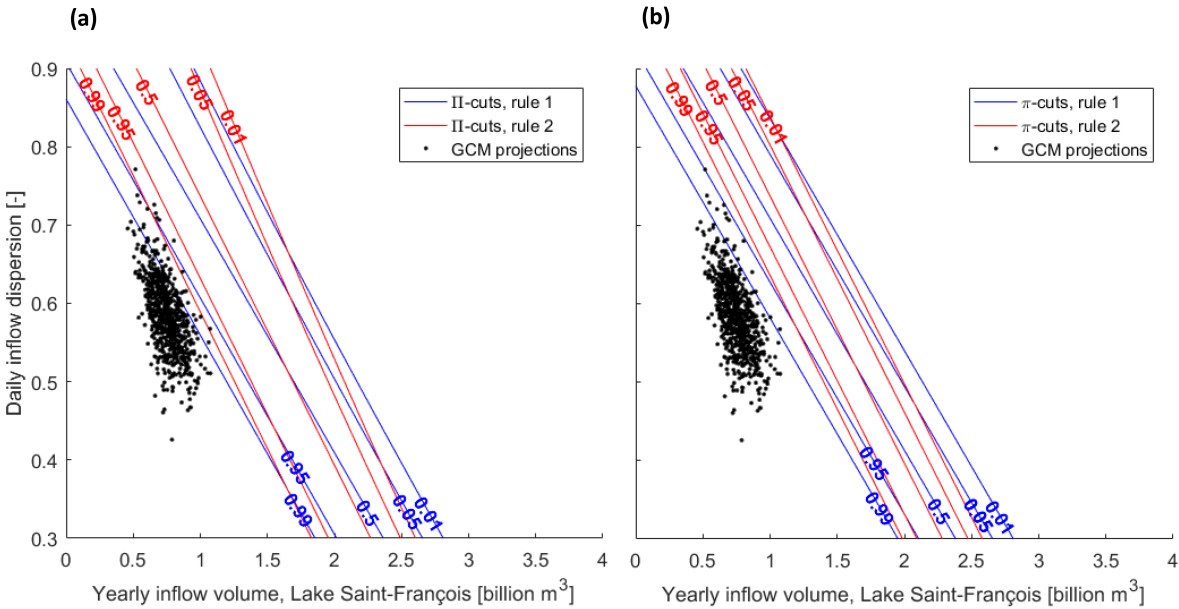

**Figure 9.** Compared logistic regressions, rules 1 and 2, and GCM projections. (a) for fuzzy outcomes (b) for binary outcomes

of this performance as acceptable or not. To integrate them in a same response surface, the methodology relies on the concept of alpha-cuts to produce an aggregated logistic regression from a membership function.

The case study of the Upper Saint-François reservoir system illustrates the implementation of the aggregated logistic regression and the conceptual framework behind it. Just like for previous uses of logistic regression, in the present method the response surface does not show a single frontier that divides the exposure space between acceptable and unacceptable flooding outcomes, but rather a parametric frontier depending, in this case, on a desired level of possibility. While possibility and probability levels cannot be directly compared (the first comprises the latter), their difference is illustrated by the wider spread of

the transition zone in the response function. This wider spread is to be expected as more sources of ambiguity are considered in the possibilistic approach, and a consequence can be that GCM projections may fall below an acceptability frontier when they do not for a probabilistic logistic regression.

   Although the main goal of this study is to propose a practical adaptation to a stakeholder-driven constraint (the absence of a clear threshold), results also explore the effect that threshold ambiguity can have on final decisions. Compared response

surfaces show that ignoring the ambiguity of a criterion can alter the comparison between options, either based on the size of acceptable domains or based on the position of GCM projections in the response surface. While applying specifically for a fuzzy approach with varying degrees of acceptability, this type of result is comparable to more general sensitivity studies over binary thresholds as in Hadjimichael et al. (2020). Not accounting for the criterion ambiguity may thus lead, in some cases and for some actors, to worse perceived floods with the selected option than with the discarded option.





Results show that the preference between options can change depending on the possibility level (a feature that that may also be found the probabilistic-only logistic regressions). When it happens, selecting the appropriate level is highly consequential and depends on the involved actors. Previous studies have linked this choice to degrees of risk aversion (Quinn et al., 2018). In the present case, the choice of the right possibility level also depends on the qualification of what degree of performance is acceptable. The ambiguity can expresses that some might consider 0.93 as acceptable, while others would only settle for 0.97.

The trade-off is thus between selecting the most robust rule for different degrees of performance. Analogous to risk aversion, this trade-off can instead be linked to loss aversion as developed by Kahneman and Tversky (1979) in prospect theory. While this study focuses on the practical integration of ambiguity as a real-world constraint, further theoretical research should focus on linking risk and loss attitudes to hydro-climatic response functions.

 The loss aversion function would be a useful concept to shape the membership function, which affects how the $\alpha$-cuts are

sampled, and thus the position of boundaries between regions in the response surface. When maximizing the range of states of the world that meet a performance criterion, selecting either a higher threshold or a lower threshold would correspond to different shapes of loss aversion functions, i.e. the weights attributed to a smaller or larger loss. A centered sigmoid shape gives more weight to the median $\alpha$-cut, corresponding to a degree of success of 0.5, and diminishing marginal improvement or loss the further the $\alpha$-cut is from the median, which is characteristic of neutral loss aversion functions, but loss-averse actors might

express a more asymmetrical membership function. Other studies have linked membership functions for fuzzy sets and prospect theory (e.g. Liu et al., 2014, Andrade et al., 2014, Gu et al., 2020).

 Defining the membership function does introduce an additional layer of complexity in the modelling process. It is ultimately up to the modeler and stakeholders to decide if it is a necessary translation of the social reality, keeping in mind how it can affect the results. The elaboration of membership functions from linguistic information is well studied in many fields (Zimmermann,

2001; Garibaldi and John, 2003; Sadollah, 2018), including in water resources management (Khazaeni et al, 2012). Future works should further explore how to elaborate adapted membership functions specific to the linguistic inputs that characterize satisfaction thresholds in the case of flood control systems, notably linking the membership function to risk and loss attitudes.

 An important caveat is that the response surface relies on a specific set of realizations from a synthetic generator and a starting data set that is perturbed and re-shuffled. The choices and assumptions that lead to a realization deserve further scrutiny in future

works. Un-gridded, on-the-fly sampling here allows exploring more freely the variability of the response, as the focus of the study is the diversity of outcomes for a given coordinate. The sampling should be improved to cover more evenly the exposure space, but without constraining too much the diversity of time series. The impact of the choice of a given synthetic generator, of the sample size and the perturbation process should be further examined. Likewise, the choice of describing variables was not the focus of the study but should be subject to an initial comparison among a larger number of candidate predictors, and

the quality of the logistic model should be further analysed and integrated as uncertainty.

 A possibilistic framework could integrate within a response surface many more of such uncertainties when probabilities are not relevant, as done in other water management studies (El-Baroudy and Simonovic, 2004, Afshar et al., 2011, Jun et al., 2013, Qiu et al., 2018, Wang et al., 2020). One particularly suitable use of fuzzy logic should be to consider as fuzzy values the ex-post expert judgement on the possibility or likeliness of the obtained synthetic time series in a given river basin. The



synthetic generator explores time series configurations but those may not always correspond to the range of outcomes expected in a watershed.

The integration of uncertainty and ambiguity quantification within the response surface tool could allow for aggregation options in a multi-objective problem (like in Poff et al., 2016, Kim et al., 2019), while easily controlling its two separate components, response uncertainty and threshold ambiguity.

Importantly, the response surface is here considered as a generic tool for decision-making under deep uncertainty, but it is used within more complex frameworks. Further research should also analyse how fuzzy thresholds can be inserted within a more complete set of methods, along with e.g. scenario discovery or adaptive approaches.

## 6   Conclusions

We explore in this study how to integrate ambiguous acceptability thresholds within uncertain response surfaces in decision-
centric vulnerability assessments. We propose a method to produce a possibilistic surface when the fuzzy threshold is applied to an uncertain surface. Aggregating logistic regressions over $\alpha$-cuts incorporates in a single measure the ambiguity of the acceptability threshold and the probability to meet such threshold, for a given state of the world. The method is illustrated with the Upper Saint-François reservoir system in Canada. We show how a fuzzy threshold shapes the response surface, and how the way this ambiguity is treated can affect the vulnerability assessment.

Challenging old probabilistic assumptions, notably in a climate crisis context, brings new tools that also imply new choices and degrees of arbitrariness. How to transparently elaborate fuzzy thresholds jointly with stakeholders, or the choice of a synthetic scenario generator, are necessary research continuations. The presented approach enables further work on stakeholder attitudes, multi-objective problems and aggregation choices. The framework here introduced to solve a practical challenge can be consolidated from a more theoretical perspective, from both possibility theory and decision making under deep uncertainty.

*Code and data availability.*   The data can be provided upon authorization from the MELCC, Québec, Canada (Ministère de l'Environnement et de la Lutte contre les Changements Climatiques). The codes required to reproduce the results are available upon request (thibaut.lachaut1@ulaval.ca).

*Author contributions.*   TL and AT conceptualized the study. TL developed the methods, models and simulations, and drafted the manuscript. AT acquired the funding and provided extensive supervision.

*Competing interests.*   The authors declare that they have no conflict of interest.



*Acknowledgements.* The work was supported by a project from Ministère de l'Environnement et de la Lutte contre les Changements Climatiques (MELCC, Québec, Canada) entitled "Étude visant l'adaptation de la gestion des barrages du système hydrique du Haut-Saint-François à l'impact des changements climatiques dans le cadre du Plan d'action 2013-2020 sur les changements climatiques (PACC 2020)". This study does not represent the views of MELCC. The authors would like to thank Louis-Guillaume Fortin, Richard Turcotte and Julie Lafleur from the CEHQ for the fruitful discussions and the knowledge of the reservoir system, Alexandre Mercille and Xavier Faucher who contributed to earlier versions of the HEC-ResSim model, and Jean-Philippe Marceau who developed the iterative Jython routine.





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
