# Peer review of "Possibilistic response surfaces: incorporating fuzzy thresholds into bottom-up flood vulnerability analysis"

_Hydrology and Earth System Sciences, 2020_

## Referee Comment (RC3)

[referee-annotated manuscript omitted]

---

## Author Response (AR1)

**Faculté des sciences et de génie**
Département de génie civil
et de génie des eaux

July 28, 2021

**Research article reference: hess-2020-646**

**Authors' response**

Dear Dr. van Griensven,

We are pleased to submit the revised version of our manuscript titled "Possibilistic response surfaces: incorporating fuzzy thresholds into bottom-up flood vulnerability analysis". We really appreciate the feedbacks provided by the three reviewers for this re-submission, and we are glad to see that the two reviewers involved in the first version of this manuscript are pleased with our changes.

We have addressed the remaining issues raised by the first reviewer. Most importantly, (1) as suggested, we have shortened and clarified the introduction: our research question is now introduced more directly and we make the distinction between the concepts of uncertainty and ambiguity; (2) we explain why we did not perform an external validation of the logistic model, emphasizing the conceptual nature of the paper, but also highlighting in the discussion the need to include an external validation should the method be fully implemented in a decision-making process.

Detailed responses to the comments of the reviewers are compiled below. We hope that you, and the three reviewers are satisfied with our responses to the comments and the revisions we made to the manuscript.

We look forward to hearing from you again to learn whether the revised paper is acceptable for publication.

Thibaut Lachaut
Amaury Tilmant, ing., PhD

Pavillon Adrien-Pouliot          418 656-2206
1065, avenue de la Médecine    Télécopieur : 418 656-2928
Local 1916                       www.gci.ulaval.ca
Québec (Québec)  G1V 0A6
CANADA

[Figure]

**Faculté des sciences et de génie**
Département de génie civil
et de génie des eaux

**Referee comment # 1**

The authors would like to thank the referee for her/his constructive and detailed comments. We reply below, with each referee's comments in bold and our response afterwards.

**General comments**

**This study was to use the fuzzy theory to refine the "bottom-up" vulnerability analysis. The authors suggest that the decision threshold itself could be ambiguous for policymakers to use the system response surface, because it would be strongly affected by stakeholders' subjective opinions. Thus, the authors proposed to incorporate fuzzy thresholds into the probabilistic response surface, and provided a case study assessing flood risks under climate change. The topic is interesting and meaningful for impact assessment research communities. To my knowledge, prior vulnerability-based assessments have been focused mostly on how to address system responses to climatic or hydrologic stressors rather than on how to set the threshold that directly determines the status of system failure. In practice, as argued by the authors, the threshold is unlikely crisp, and hence its ambiguity needs to be condensed in the response surface. I think the authors' quantitative approach is novel and reasonable, and recommend minor revisions for final publication in Hydrology and Earth System Sciences. Please find my specific comments below.**

- **I recommend to distinguish ambiguity of the threshold from modeling uncertainty through the manuscript. While ambiguity and uncertainty are similar and thus some authors often lump the two concepts, sources of the two seem different in the bottom-up assessment. In a climate change impact assessment, typical uncertainty sources are uncertain future emissions, uncertain general circulation models, uncertain system and hydrologic models, i.e., mostly from things outside of humans' psychological behaviors. On the other hand, a major source of the ambiguity would be stakeholders' subjective opinions about the decision threshold, which are likely affected by their sub conscious behaviors. If uncertainty is treated as a modeling problem and separated from the fuzziness of the threshold in the manuscript, readers could understand the authors' intention more clearly. I do not mean that uncertainty cannot be lumped with ambiguity, but separating the two would better guide potential readers. The manuscript is starting with "Uncertainty is a defining feature of water management." Uncertainty seems to include ambiguity in the manuscript from the beginning.**

Thank you for your comment. We propose the following changes in the manuscript to make the distinction between uncertainty (on the modelling side) and ambiguity (on the stakeholder side). We replace "uncertainty" by "imperfect knowledge" whenever the word is used as

Pavillon Adrien-Pouliot
1065, avenue de la Médecine
Local 1916
Québec (Québec)  G1V 0A6
CANADA

418 656-2206
Télécopieur : 418 656-2928
www.gci.ulaval.ca

generic term that lumps together both hydro-climatic uncertainty and ambiguity. Added text is underlined, removed text is strikethrough.

L8, abstract: modified. "Incorporating fuzzy thresholds therefore requires articulating  categories of imperfect knowledge that are different in nature…"

L18. First paragraph changed: "Imperfect knowledge is a defining feature of water resources management. For example, the uncertainty about the availability of water at any given time drives the development of storage capacities."

L97 to L101: modified. "The present study articulates two  categories of imperfect knowledge: the ambiguity of the acceptability threshold and the uncertainty of the response surface.

L149: modified. "non-probabilistic  imperfect knowledge"

L198: modified. "…we introduce a  quantification of ambiguity that is different in nature from the irreducible hydro-climatic and modelling uncertainty"

L397: modified. "… incorporate  the ambiguity of…"

L402: "The two sources of uncertainty are different in nature" replaced by "These two concepts represent different sources of imperfect knowledge".

- **Please shorten the introduction. To me, it was too long. For example, the sentences about top-down assessments is not the core of this work. Just introducing some shortcomings and leaving some relevant references would be better to lead readers directly to the main objectives of this work.**

We have shortened the introduction and presented our research question more quickly.

- **A remaining task in the case study might be to validate the probability estimates from the logistic regression. Kim et al. (2019) handled this problem by additional simulations with random climatic stresses independent of the logistic regression. Please consider any method that could show validity of the probability estimates. Obviously, it will improve reliability of this work.**

We agree that adding an external validation process would strengthen the illustration made with the St-François case study. However, the article is intended more as a proof of concept about the inclusion of fuzzy sets within a logistic regression method (illustrated with a real problem) rather than a ready-to-use decision framework. As in Quinn et al. (2018) and

[Figure]

**Faculté des sciences et de génie**
Département de génie civil
et de génie des eaux

Hadjimichael et al. (2020), we focus on other aspects of the logistic regression, without producing an external validation for now.

We also mention several other steps that would be required to move from a proof-of-concept to a reliable recommendation for the reservoir system, such as the elicitation of the membership function or the selection of the most influential stressors. The external validation is now included in that list. Besides, further studies should integrate the error stemming from external validation in a possibilistic manner, along with the quality of fit ($R^2$).

L449-450, rephrased and expanded: "Finally, the quality of the logistic model should be further analysed. External validation with a separate sampling of the exposure space should be included (Kim et al., 2019). Further work should seek to integrate the goodness of fit and the external validation as additional sources of uncertainty within the method. All the aforementioned steps should be considered for this possibilistic method to be used as policy recommendation."

- **I think the alpha-cut approach has an issue of how to set the alpha value appropriately, as though the traditional logistic regression has an issue of how to appropriately set the pi threshold. The authors need to leave some discussion on this issue.**

With the presented method, we would say that the choice is on the Π threshold (possibility) rather than on the alpha-cut only. The alpha-cut serves as a mapping to aggregate the logistic regressions. Selecting an alpha-cut would remove the fuzziness from the problem. However, selecting the right Π – possibility – value remains a challenge (just like the choice of the adequate membership function that maps the alpha values to the alpha cuts).

. To highlight this question, we propose the following change:

L420: "Results show that the preference between options can change depending on the possibility level Π. . When it happens, selecting the appropriate  Π threshold is highly consequential and depends on the involved actors. This challenge is the equivalent in possibilistic terms to the selection of the probability threshold pi in the non-fuzzy logistic regression (Kim et al., 2019). The Π threshold depends on both a probability level and the value of α."

**Following is line by line comments on technical errors and some issues on the authors' discussion.**

**L5: please remove the duplicated "the"**

[Figure]

**Faculté des sciences et de génie**
Département de génie civil
et de génie des eaux

Done. Thank you.

**L45-46: Please explain what the climate impact response function first. Then, use the acronym. Perhaps, a relationship between such stressors and the performance of the system in L45 is the CIRF. The reverse CIRF might be used to find the range of climate stressors within which system performance is acceptable. I feel that this part needs clearer explanation.**

The mention of the CIRF has been removed to streamline the introduction, references are kept among the studies using some form of response function.

**L53-56: This sentence is too long. Please consider rewriting.**

The sentence is removed from the introduction and re-written in section 2, L105.

"In the Decision Scaling approach (Brown et al., 2012, Poff et al., 2016, Brown et al., 2019) GCM projections can  be introduced as weights on the response surface to inform probabilities associated to climate states. GCMs can thus remain useful without conditioning the decision process. Their weights can be updated as uncertainty is resolved, resulting in a refined estimate of the expected system outcome over the response surface ."

**L79-96: Some of this part could be moved to the section 2, because it includes how fuzzy theory is applied in this work.**

L84 to 94: moved to section 2, L131.
L83, added: "there is interest in integrating estimates of uncertainty into the response surface tool, as performed recently through logistic regression (Quinn et al., 2018, Kim et al., 2019)."

**L128: With two variables**

Done.

**L130: just underestimate (no hyphen)**

Done.

**L147: Zadeh (1965)**

Done.

**L163: Please check if the inequality symbol is correct in Eq.(3)**

Pavillon Adrien-Pouliot            418 656-2206
1065, avenue de la Médecine        Télécopieur : 418 656-2928
Local 1916                         www.gci.ulaval.ca
Québec (Québec)  G1V 0A6
CANADA

[Figure]

**Faculté des sciences et de génie**
Département de génie civil
et de génie des eaux

Corrected, thank you. Second line is now $\theta_1 \leq R < \theta_2$

**L173: … threshold, showing …**

Done.

**L185: if you chose just two explanatory variables (x1, x2), then the eq. 5 should have x1 and x2. Please remove "+…"**

Corrected.

**L240: Please be consistent with the citation policy of the journal. Nazemi et al. (2013), Borgomeo et al. (2015), …**

Done.

**L242: Here too. Vormoor et al. (2017)**

Done.

**L248-256: Could you add any statistics resulted from the bias correction methods? It could inform reproducibility of CMIP5 GCMs. I guess runoff projections were likely used for those bias-corrected projections. Then, the scale and boundary mismatch between the GCMs and the watersheds are still a problem. If I am correct, please leave discussion on this issue in the manuscript. If not, please more clearly describe the inflow projections were made.**

Those statistics are unfortunately not directly available. Hydrologic projections were provided by the government agency responsible for producing a province-wide analysis of the alteration of flow regimes due to climate change. To achieve this, they developed a large-scale modelling platform to automatically analyze the Southern part of the Quebec province ($> 7000.000$ km$^2$). Meteorological series were bias-corrected for the reference climate and then processed by the Hydrotel model for all major rivers in the Southern part of the Quebec province. For our case study, the resulting hydrologic simulations were also bias-corrected with the historical flow record and the quantile mapping approach.

L248 to 258, modified: "Streamflow scenarios are provided by MELCC through the Quebec Water Atlas 2015 (CEHQ, 2015, MELCC, 2018). Those hydrologic projections are based on climatic projections from the Natural Resources Canada data base of GCM simulations (CMIP5, Hydro-climatic Atlas, 2015)  Meteorological time series were bias-corrected by the Québec Water Agency for the reference climate (1971-2000 period) and then processed by the Hydrotel model (Fortin, 2001) for all major rivers in the Southern part of the Quebec province. For the Upper Saint-François River

Pavillon Adrien-Pouliot          418 656-2206
1065, avenue de la Médecine      Télécopieur : 418 656-2928
Local 1916                       www.gci.ulaval.ca
Québec (Québec)  G1V 0A6
CANADA

[Figure]

**Faculté des sciences et de génie**
Département de génie civil
et de génie des eaux

Basin, resulting hydrological simulations were also bias-corrected with the historical flow record using the quantile mapping approach."

**L315: 5-year period**

Done.

**L336: please remove the unnecessary a.**

Done.

**L340: The pseudo R2 is about 75%. Is it acceptable performance? And, what are potential sources of the remaining 25%? Please add the authors' opinion on this result**.

We indeed add our opinion on the $R^2$ value and options to improve it.

L342, added: "These values are considered as an acceptable goodness of fit for this study. A pseudo $R^2$ equal to 1 represents a perfect model, and a value of 0 means the logistic model is not a better predictor of probabilities than an intercept-only model. A possible room for improvement of the predicting value of the model would be to change the predictors, although it was not the core of this study. Selecting two different predictors from a larger set of candidates might increase the final $R^2$ (performing a first round of logistic regressions for each pair and selecting the pair with highest $R^2$ as in Quinn et al., 2018)."

**L420-428: Maybe, this part is related to how to set the alpha threshold. Am I right?**

Indeed. Addressing both this remark and the 4[th] point of the main comments, the paragraph is modified as follows:

L420: "Results show that the preference between options can change depending on the possibility level Pi. . When it happens, selecting the appropriate  Pi threshold is highly consequential and depends on the involved actors. This challenge is the equivalent in possibilistic terms to the selection of the probability threshold pi in the non-fuzzy logistic regression (Kim et al., 2019). The Pi threshold depends on both a probability level and the value of alpha."

Pavillon Adrien-Pouliot
1065, avenue de la Médecine
Local 1916
Québec (Québec) G1V 0A6
CANADA

418 656-2206
Télécopieur : 418 656-2928
www.gci.ulaval.ca

[Figure]

**Faculté des sciences et de génie**
Département de génie civil
et de génie des eaux

**L429-436: The introduction of the loss aversion function and the membership function is somehow abrupt. Please explain those concepts first.**

The membership function is explained in section 2, a letter µ should be added here for clarity. The paragraph (L429- L436) establishing a relation between membership function and loss aversion actually responds to the concern from another reviewer in the previous version, about how to elicit the membership function in practice. The loss aversion concept can be a suggestion for future work in that regard. For clarity, this paragraph is thus swapped with the next paragraph (L437-L442), which addresses the challenge of finding a membership function. The logical order should be : (i) difficulty of setting the Pi threshold (ii) difficulty of elaborating the membership function µ (iii) loss aversion (prospect theory) as a possible response in future research. Based on this comment, we also believe the mention of prospect theory can be shorter, as it is only a suggestion for future research.

We start from the previously mentioned change (L420):

L420: "[…] The Π (possibility) threshold depends on both a probability level and the value of alpha. The present possibilistic framework introduces a potential trade-off when selecting robust alternatives for different degrees of acceptability. "

(moved up) "Defining the membership function µ introduces an additional layer of complexity in the modelling process. It is a determining step as it defines the position of the Π thresholds in the exposure space. It is ultimately up to the modeler and stakeholders to decide if the fuzzy set is a necessary translation […] in the case of flood control systems."

To further address both challenges of selecting the appropriate Π level and eliciting the membership function µ, loss aversion, as developed by Kahneman and Tversky (1979) in prospect theory, would also be a useful concept. A parallel can be drawn with Quinn et al. (2018), where the choice of the probability level π in a logistic regression is instead linked to risk aversion.

 Other studies have linked prospect theory with membership functions for fuzzy sets (e.g. Liu et al., 2014, Andrade et al., 2014, Gu et al., 2020). While this study focuses on the practical integration of ambiguity as a real-world constraint, further theoretical research should focus on linking both risk and loss attitudes to hydro-climatic response functions.

**L473: The proposed method is probably one of various ways to consider fuzziness of the decision threshold in the bottom-up impact assessment framework. We don't know yet whether or not it works for other problems with very high complexity. Please tone down at least somehow.**

[Figure]

We agree the sentence is misleading and is now changed:

L473: "The presented approach  can be followed by further work on stakeholder attitudes, multi-objective problems and aggregation choices in bottom-up vulnerability assessments. The framework here introduced to solve a practical challenge can be consolidated from a more theoretical perspective, from both possibility theory and decision making under deep uncertainty."

**Referee comment #2:**

**The authors have provided a substantial revision. The results are much sharper now, and the literature review has been expanded. The use case for this method is still not fully clear, with the difficulty of specifying the membership function. However, I believe the value in the paper is the mathematical foundation of how to apply fuzzy set theory to bottom-up decision making, if and when stakeholders could identify such a function. I recommend that the paper be accepted in present form.**

We would like to thank the referee for his/her constructive comments which have led to the improvement of the paper.

**Referee comment #3:**

**I think the authors have done a good job reducing the scope of this paper, focusing on its main contribution. I have only minor editorial suggestions in the attached manuscript that the authors might consider before publication.**

We would like to thank the referee for this second review. We are grateful for the detailed corrections, and we have applied all the changes suggested in the supplementary material in the final version. There is only one clarification that we would like to make: by the end of the introduction, we do consider that the internal uncertainty of the response function is the challenge to the application of a fuzzy threshold, not the motivation. The sentence could be clearer, so we propose this change:

Previous version, L95: "Such internal uncertainty to the response function challenges the introduction of fuzzy thresholds, as the separation of the exposure space into acceptable and unacceptable regions is not obvious even with a binary definition of acceptability."

Modified version: "These studies show that, even with a crisp acceptability threshold, the internal uncertainty of the response surface can challenge the separation of the exposure

[Figure]

**Faculté des sciences et de génie**
Département de génie civil
et de génie des eaux

space. Introducing a fuzzy threshold to a response surface that also has its own uncertainty is not trivial as these concepts address forms of imperfect knowledge that are very different in nature."

**Minor corrections and changes from the authors:**

L80: Subtitle 2.1.1. modified: "Uncertain response  surfaces"

L121: "R" instead of "p".

L182: the subtitle 2.2.1 is removed, as not needed anymore.

Fig 5: missing map reference added (MELCC, 2018)

Added references for completeness: Fortin et al. (2007), L234, Loucks and Beek (2017), L105, L297; Le Cozannet et al. (2017), L56, 459.

Pavillon Adrien-Pouliot          418 656-2206
1065, avenue de la Médecine      Télécopieur : 418 656-2928
Local 1916                       www.gci.ulaval.ca
Québec (Québec)  G1V 0A6
CANADA